# Novel Plant-Protein (Quinoa) Derived Bioactive Peptides with Potential Anti-Hypercholesterolemic Activities: Identification, Characterization and Molecular Docking of Bioactive Peptides

**DOI:** 10.3390/foods12061327

**Published:** 2023-03-20

**Authors:** Feyisola Fisayo Ajayi, Priti Mudgil, Amie Jobe, Priya Antony, Ranjit Vijayan, Chee-Yuen Gan, Sajid Maqsood

**Affiliations:** 1Department of Food Science, College of Agriculture and Veterinary Medicine, United Arab Emirates University, Al-Ain 15551, United Arab Emirates; 202190079@uaeu.ac.ae (F.F.A.); priti.d@uaeu.ac.ae (P.M.); 2Department of Biology, College of Science, United Arab Emirates University, Al-Ain 15551, United Arab Emirates; 201450107@uaeu.ac.ae (A.J.); 201990021@uaeu.ac.ae (P.A.); ranjit.v@uaeu.ac.ae (R.V.); 3The Big Data Analytics Center, United Arab Emirates University, Al-Ain 15551, United Arab Emirates; 4Zayed Center for Health Sciences, United Arab Emirates University, Al-Ain 15551, United Arab Emirates; 5Analytical Biochemistry Research Centre (ABrC), Universiti Sains Malaysia, SAINS@USM Campus, Bayan Lepas 11900, Malaysia; cygan@usm.my

**Keywords:** quinoa protein, protein hydrolysate, bioactive peptides, cholesterol esterase, pancreatic lipase

## Abstract

Hypercholesterolemia remains a serious global public health concern. Previously, synthetic anti-hypercholesterolemic drugs were used for ameliorating this condition; however, long-term usage presented several side-effects. In this regard, natural products as an adjunct therapy has emerged in recent times. This study aimed to produce novel bioactive peptides with anti-hypercholesterolemic activity (cholesterol esterase (CEase) and pancreatic lipase (PL)) from quinoa protein hydrolysates (QPHs) using three enzymatic hydrolysis methods (chymotrypsin, protease and bromelain) at 2-h hydrolysis intervals (2, 4, and 6 h). Chymotrypsin-generated hydrolysates showed higher CEase (IC_50_: 0.51 mg/mL at 2 h) and PL (IC_50_: 0.78 mg/mL at 6 h) inhibitory potential in comparison to other derived hydrolysates and intact quinoa proteins. Peptide profiling by LC-MS QTOF and in silico interaction with target enzymes showed that only four derived bioactive peptides from QPHs could bind in the active site of CEase, whereas twelve peptides could bind in the active site of PL. Peptides QHPHGLGALCAAPPST, HVQGHPALPGVPAHW, and ASNLDNPSPEGTVM were identified to be potential CEase inhibitors, and FSAGGLP, QHPHGLGALCAAPPST, KIVLDSDDPLFGGF, MFVPVPH, and HVQGHPALPGVPAHW were identified as potential PL inhibitors on the basis of the maximum number of reactive residues in these bioactive peptides. In conclusion, QPHs can be considered as an alternative therapy for the treatment of hypercholesterolemia.

## 1. Introduction

The rise in diet-related metabolic disorders such as hyperlipidemic or hypercholesterolemia, cardiovascular diseases, hypertension, and other related metabolic diseases has been identified as the major cause of mortality globally, and thus has become a serious public health issue. The consumption of processed food and animal products has been associated with the occurrences of these health issues. Cholesterol esters are normally found in the human diet, which the human intestine cannot sufficiently absorb, thereby producing free fatty acids and cholesterol that elevate the level of blood cholesterol. It is a known fact that rising levels of cholesterol and triglycerides in the blood serum results in hyperlipidemia or hypercholesterolemia [1]. According to [2], the molecular mechanism for understanding the physiological development of hyperlipidemia is quite difficult. However, two enzymes (cholesterol esterase (CEase) and pancreatic lipase (PL) have been associated with hyperlipidemia and hypercholesterolemia complications, and their inhibitory potentials have been targeted by scientists in order to develop anti-hypercholesteremic drugs [1,3]. For many years, synthetic drugs have been used to lower blood cholesterol production and absorption. However, synthetic drugs (mainly the CEase or PL inhibitors) are often accompanied with harmful consequences on human health and metabolism due to long-term usage. Therefore, there is a high demand for natural products that can prevent these malignant diseases.

Nowadays, generating bioactive peptides from plant-based proteins have gained more attention owing to their biological and functional activities, which is highly significant in alleviating chronic diseases, as well as in promoting human health. Moreover, the effectiveness of the derived peptides depends on their sequence, structure, and inherent amino acid composition. Quinoa proteins are a very good source of essential amino acids [4]. Apart from providing significant nutrients, they also possess several peptide sequences with specific physiological benefits [5]. Protein hydrolysates from quinoa have been reported to be a potential ingredient in designing functional foods and nutraceuticals because of their multifunctional bioactivity, including antioxidant, antihypertensive, antidiabetic, dipeptidyl peptidase IV (DPP-IV) inhibitory, and angiotensin-converting enzyme (ACE) inhibitory properties [6,7].

Similarly, several bioactive peptides from protein hydrolysates have demonstrated the capacity to inhibit the digestion of dietary lipids in in-vitro and cultured cell assays [8]. Peptides that possess anti-hyperlipidemic activity operate in the gastrointestinal tract, hepatocytes, and adipocytes by binding cholesterol micelles and bile acids, thereby hindering their absorption. According to Boachie, Yao and Udenigwe [8], their operation affects the activity of some regulatory enzymes, particularly the 3-hydroxy-3-methylglutaryl CoA reductase (HMGCoAR), which is well-known to be primarily responsible for cholesterol biosynthesis, and this enzyme inhibition is targeted to decrease the rate of cholesterol biosynthesis.

Several scientific investigations have shown that natural peptides found in plant proteins can inhibit cholesterol absorption or HMGCoAR activity [9]. However, protein hydrolysates and bioactive peptides derived from quinoa have not been studied for their anti-hypercholesterolemic effect via the inhibition of CEase and PL enzymes to date. Therefore, this study investigated the role and binding mechanism of quinoa derived bioactive peptides for inhibiting enzymatic markers i.e., CEase and PL that are responsible for hypercholesterolemia. For effective inhibition of the cholesterol metabolizing enzymes, the potential of generated bioactive peptides to bind to their specific hotspots on the active site of the enzyme depends on the cleavage sites where the proteolytic enzymes will act on the protein [10]. As such, different enzymes can result in generating varying amino acids sequences in peptides, which is vital in demonstrating their bioactivity. Moreover, numerous studies have demonstrated the potential of anti-hypercholesterolemic peptides derived using specific enzymes (alcalase, pepsin, trypsin, flavourzyme, chymotrypsin, etc.) in inhibiting CEase and PL active sites [11,12,13].

Accordingly, quinoa protein isolates were hydrolyzed using three different enzymes (chymotrypsin, protease, and bromelain) in order to get a diverse sequence of peptides which might be potential inhibitors of CEase and PL. The generated quinoa hydrolysates were characterized biochemically via the degree of hydrolysis, and their CEase and PL inhibitory potentials were determined. Furthermore, peptides generated from quinoa protein isolates were characterized and identified using LC-MS-QTOF. A molecular docking approach was carried out to elucidate the molecular binding mechanism between potent anti-hypercholesterolemia peptides with the target enzymes (CE and PL) through the in silico structural activity relationship.

## 2. Materials and Methods

### 2.1. Materials (Quinoa Seeds) and Chemicals

Quinoa seeds used in this study were purchased from the local market (Al-Ain, United Arab Emirates). Enzymes such as protease (Type XIV; ≥3.5 units/mg solid derived from *Streptomyces griseus*), α-chymotrypsin (EC 3.4.21.1 from bovine pancreas; C4129 40 units/mg), and bromelain (B4882: 3–7 units/mg solid; EC 3.4.22.33 from pineapple stem), HPLC grade reagents (acetonitrile and methanol), and other chemicals such as formic acid *p*-nitrophenyl butyrate, *o*-phthaldialdehyde, trizma base, β-mercaptoethanol, SDS, and sodium tetra-borate were procured from Sigma Aldrich (St. Louis, MO, USA). Other analytical grade reagents used were purchased from the UAE (BDH Middle East, Dubai, United Arab Emirates).

### 2.2. Preparation of Quinoa Protein Isolate

Firstly, quinoa seeds were oven dried (40 °C for 3 h), dry-blended (IKA A11, Guangzhou, China), and passed through a screen (3 μm pore size) to obtain a fine powder. The methodology described by [6] as adopted from [14] was used to obtain quinoa protein isolate (QPI). In sum, defatted quinoa flour was dispersed in 0.015 mol L^−1^ NaOH (1:10 ratio) maintained at a pH of 10–11 and was stirred for 2 h on an electric shaker (24 ± 2 °C), and the resulting slurry was stored at 4 °C overnight. The slurry was homogenized for consistency and centrifuged twice at 4 °C at a speed of 10,000× *g* for 20 min after overnight storage. Whatman No. 1 paper was used to filter the obtained supernatant, and the pH of the filtrate reduced to 4.5 using the pH drop method with 1.0 mol L^−1^ HCl solution to precipitate quinoa protein. Afterwards, precipitated proteins were separated by centrifuging at 4 °C for 10 min at the speed of 10,000× *g*, and were subsequently washed twice with deionized water. The pH of the resultant pellets was adjusted to pH 7.0 (with 1.0 mol L^−1^ NaOH) and stored at −20 °C until further use. The Kjeldahl method was adopted to determine the protein content of quinoa protein isolate (QPI) slurries, which was found to be 44%.

### 2.3. Quinoa Protein Hydrolysate Production

The methodology as described by [6] was used in the production of quinoa protein hydrolysates (QPHs). The obtained QPI slurry was thoroughly homogenized (T 25 digital ULTRA-TURRAX^®^ homogenizer IKA^®^-Werke GmbH, Staufen, Germany). The protein content of the slurry was set to 4% with deionized water, and the content was divided into four batches for treatment with bromelain, chymotrypsin, and protease, respectively, while the last part was kept as control. Then, 1 mol/L NaOH and 5N HCl were used to adjust the pH of the treated protein slurries for the bromelain enzyme to 7.0, the chymotrypsin enzyme to 7.8, and the protease enzyme to 8.0, respectively. Furthermore, enzyme concentration was pre-calculated, suspended in 1 mL of distilled water resulting in a 1% enzyme/substrate ratio, and subsequently added to the QPI slurry. The samples were then distributed into 50 mL falcon tubes in triplicate for each period of hydrolysis and incubated (water bath) for up to 6 h at the speed of 100 ramps/min at a controlled temperature of 50 °C. Samples were taken every 2 h from all enzymatic incubations, and the reaction was inactivated at 95 °C for 600 s. Generated QPHs were centrifuged at 4 °C at a speed of 10,000× *g* for 10 min, and supernatants were collected and stored at −20 °C for further analysis. All of the analyses were performed within two weeks of the production of hydrolysates.

### 2.4. Degree of Hydrolysis (DH%)

The previously published o-phthaldialdehyde (OPA) methodology of [15] was used for the determination of DH values.
(1)$$Degree\;of\;hydrolysis\left(DH\%\right)=\frac{h}{h_{tot}}\times 100$$

Herein, *h_tot_* = total number of peptide bonds per protein equivalent; and *h* = number of hydrolyzed bonds, and calculated using *h* = (SerineNH2 − β)/α. Where *h_tot_*, β, and α, β and *h_tot_* values for quinoa proteins were used as 8.0, 0.40, and 1.00 mEq/g protein, respectively.

### 2.5. Anti-hypercholesterolemia Assays

#### 2.5.1. Determination of Pancreatic Lipase (PL) Inhibitory Activity of QPHs

The PL-inhibitory activity of generated hydrolysates was measured using the methodology of [13]. Here, PL (20 μL) and *p*-nitrophenyl butyrate (25 μL) were mixed with each sample (50 μL) in a sodium chloride: sodium phosphate buffer (pH 7.2, 100 mM), and then incubated in a 96-well microplate reader. The quantity of the resultant reaction was modified with the above buffer to 150 μL and incubated (37 °C for 30 min). After the incubation period, the released *p*-nitrophenyl for each sample was measured on a plate reader (Epoch 2, BioTek, Winooski, VT, USA) at 405 nm. The PL-inhibitory activity was calculated using the equation below
(2)$$\%\;Enzyme\;inhibition=\left(1-\left(\frac{C-D}{A-B}\right)\right)\times 100$$

Here: *A* = the absorbance of control; *B* = the absorbance of control reaction blank; *C* = the absorbance of sample, and *D* = the absorbance of the sample blank. The IC_50_ values were determined by plotting the percentage inhibition against the concentration of the test compound and expressed in mg (eqv protein)/mL.

#### 2.5.2. Determination of Cholesterol Esterase Inhibitory Activity of QPHs

The CEase-inhibitory activity was determined as per the procedure documented by [13]. Briefly, QPHs (25 μL), substrate (50 μL) containing 5 mM *p*-nitrophenyl butyrate in sodium phosphate (100 mM): NaCl buffer (pH 7.2, 100 mM) was placed in a 96-well microtiter plate. Subsequently, the mixtures were incubated with 50 μL of porcine pancreatic CEase (5 μg/mL) at 37 °C for 180 min. After that, the *p*-nitrophenol released from the enzymatic hydrolysis of *p*-nitrophenyl butyrate was read at 405 nm. The percentage CEase inhibition was determined using Equation (2) above. The IC_50_ values were determined by plotting the percentage inhibition against the concentration of the test compound and expressed as mg (eqv protein)/mL.

### 2.6. Peptide Identification

#### 2.6.1. Sequencing of Peptides Implied in PL and CEase-Inhibitory Activities Using Liquid Chromatography-Mass Spectrometry Quadrupole Time-of-Flight (LC-MS Q T-O-F)

For the identification of peptides in selected hydrolysates, QC-6 (chymotrypsin-generated at 6 h hydrolysis), LC-MS Q T-O-F was conducted as described previously by [16]. The identification of peptides was performed as follows: Advance Bio Peptide Map, C18 column (2.1 × 100 mm, 2.7 µm particles; Agilent, Santa Clara, CA, USA) was used for the peptide separation whereas the LC-MS Q T-O-F (model 6520, Agilent, Santa Clara, CA, USA) was used for the analysis. The mobile phases used were: (A) deionized water containing 0.1% formic acid, and (B) acetonitrile containing 0.1% (*v*/*v*) formic acid with a flow rate of 15 µL/min. The mobile phase gradient of the HPLC system was as follows: (a) 0–5 min, 10% B; (b) 5–115 min, 10–95% B, (c) 115–120 min, 95% B, (d) 120–135 min, 95–10% B, and (e) 140–150 min, 10% B. The electrospray ionization-quadrupole-time-of flight system (ESI-QTOF) condition was: (a) mass range: 70–2000 m/z; (b) collision energy: 6V/100 Da (offset-2); (c) flow rate: 15 µL/min; (d) ion spray sources: 3.5 kV; (e) drying gas: Nitrogen at a temperature of 350 °C with a flow rate of 10 L/min; (f) nebulizer pressure: 3 psig; (g) fragmentor voltage: 110 V; and (h) fragmentation mode: collision induced dissociation (CID). Data generated from the mass spectrometry approach were analyzed using PEAKS studio version 6.0 [17]. The precursor was selected on the basis of a minimum charge of 2 and a maximum charge of 10. From the generated peptides, only peptides with average local confidence (ALC) > 70% were chosen for further analysis.

#### 2.6.2. Potential Biologically Active Peptides Selection Using PeptideRanker

Potential bioactive peptides were screened using the PeptideRanker web server (http://bioware.ucd.ie/, accessed on 26 July 2022) [18]. Peptides showing a score of more than 0.5 were regarded as potentially biologically active and subjected to further in silico analysis.

#### 2.6.3. Selection of Identified PL and CEase Inhibitory Peptides Using Peptide Ranker

The preliminary insight into the molecular mechanism of peptides for their PL and CEase inhibitory activities were explored through an in silico docking investigation using the online program Pepsite2, accessible at http://pepsite2.russellab.org (accessed on 16–19 August 2022) [19]. Herein, 3D structures of CEase (PDB code: 1AQL) and PL (PDB code: 1ETH) were introduced from the Protein Data Bank through http://www.rcsb.org/pdb/ (accessed on 16–19 August 2022). Peptide inputs were then entered jointly with a protein receptor in PDB format. The best interaction of peptides with enzymes showing *p*-values < 0.05 and the interaction with key hotspots on enzymes playing a significant role in enzyme inhibition were taken into consideration for further analysis.

### 2.7. Molecular Docking

#### 2.7.1. Preparation of Protein Structure

The 3D structures of human and bovine CEase, PDB IDs 1F6W and 1AQL respectively, and PL (PDB IDs: 1LPB and 1ETH) were obtained from the Protein Data Bank (PDB) [20]. These proteins were processed and optimized prior to docking using the Schrödinger Suite’s Protein Preparation Wizard (Schrödinger Suite 2021-1: Protein Preparation Wizard; Schrödinger LLC, New York, NY, USA). This workflow prepares a PDB structure for docking by adding and optimizing hydrogen bonds, assigning bond orders, simplifying multimeric complexes, creating disulfide bonds, deleting unwanted water molecules, adjusting ionization states, fixing disoriented groups, filling missing loops and sidechains and finally, energy minimizing and optimizing to produce a geometrically stable structure [21].

#### 2.7.2. Active Site Identification and Grid Generation

The active sites of CEase (PDB IDs: 1F6 W and 1AQL) and PL (PDB IDs: 1LPB and 1ETH) that are directly involved in ligand-binding were identified from the literature The active site of lipase is located in the N-terminal domain and their catalytic activity is mediated by the catalytic triad Ser152, Asp176 and His263 Similarly, CEase also shares a comparable catalytic triad composed of Ser194, Asp320 and His435. For the docking analysis, a receptor grid suitable for peptide docking was generated for the minimized protein structures enveloping the active site residues. No constraints were applied when default parameters for van der Waals scaling factor (1.00) and charge cut-off (0.25) were used. The OPLS 2005 force field was used for structural representation [22].

#### 2.7.3. Peptide Docking and Binding Free Energy Calculation

The Peptide Docking panel of Schrödinger Maestro [23] was employed to perform the docking of peptides to the protein structures. Using this tool, small peptides lower than 16 amino acids can be docked and scored with either Glide Score or molecular mechanics-generalized Born surface area (MM-GBSA) approach. Each peptide was docked to the protein receptor using Glide with increased sampling in several docking runs. Subsequently, OPLS molecular mechanics force field was used for pose optimization. After carrying out conformer clustering, ten representative peptide poses were chosen. Lastly, selected poses were re-scored and ranked based on GlideScore empirical scoring function [23,24].

The binding free energy of the best docked poses obtained after peptide docking were evaluated using the MM-GBSA approach. Binding free energy calculations were performed using Schrödinger Prime with OPLS 2005 force field combined with VSGB 2.0 implicit solvent model. The peptide was minimized, and the receptor was treated as rigid for the MM-GBSA calculations.

### 2.8. Statistical Analysis

All hydrolysates were produced in three batches representing three replicates. All analyses were carried out in triplicate. Data analysis was performed with SPSS 24.0 statistical software (SPSS INC., Chicago, IL, USA) using one-way analysis of variance. Means separation was conducted using Tukey’s multiple range test, and a *p*-value of 5% was defined as statistically significant (*p* < 0.05).

## 3. Results & Discussion

### 3.1. Degree of Hydrolysis (DH%)

The DH and enzyme specificity are among the measurable factors to be controlled because they define the biological activities of the resulting peptides. Studying the hydrolysis of protein reveals the capability of an enzyme to degrade proteinaceous substrate, and this serves as a pointer for their proteolytic reaction [25]. Noteworthy, the protein profile obtained from intact quinoa (QPI), and quinoa protein hydrolysates (QPHs) from the three enzymes understudied after 2, 4, and 6 h of hydrolysis is depicted in Appendix A.

In this work, the effects of different enzymes (chymotrypsin, protease, and bromelain) at 2 h-interval reaction periods on the DH of quinoa protein hydrolysates (QPHs) were reported (Table 1).

The results obtained showed a characteristic similar hydrolysis pattern among all the enzymes, where the DH values significantly increased with the progression of hydrolysis time up to 6 h (*p* < 0.05). These results are comparable with those obtained by [26], where they reported increased DH values of quinoa protein hydrolysates as a function of increased hydrolysis time. Moreover, varying DH values were demonstrated among the QPHs generated using the three enzymes of study. The DH levels in the range of 36.01 to 66.05%, and 45.05 to 63.3%, and 34.99 to 51.91% were obtained for chymotrypsin, protease, and bromelain generated QPHs, respectively.

Based on the enhanced DH values with progressive hydrolysis time, chymotrypsin-produced hydrolysates after 6 h (QC-6) displayed the highest value of 66.05%. However, it was not noted that the DH value obtained in protease-produced hydrolysates after 6 h (QP-6) was not significantly different from the DH value of QC-6h (*p* > 0.05). Bromelain-produced hydrolysates after 6 h (QB-6) showed an intermediate DH value of 51.91%, and the lowest protein hydrolysis was demonstrated in both bromelain-produced hydrolysates (QB-2) and chymotrypsin-produced (QC-2) after 2 h of bio-catalysis (*p* > 0.05). In general, chymotrypsin was found to be the most effective enzyme in hydrolyzing quinoa proteins as opposed to the other two proteolytic enzymes (bromelain and protease). The hydrolytic pattern of chymotrypsin indicates greater proteolytic efficiency of the enzyme, which may be attributed to the size and action of the released peptides. As a result, the abundance of small-sized peptides with enhanced effectiveness against metabolic markers could be produced from chymotrypsin at prolonged hydrolysis periods. This is consistent with previous studies that suggested that higher DH due to disintegration of more peptide bonds could be attributed to the higher activity of the enzyme [27].

The enzymatic hydrolysis of quinoa proteins using bromelain and chymotrypsin exhibited a rapid and steady increase in the DH values over a period of 6 h, an indication that the larger peptides that were released at a lower hydrolysis time were further disintegrated into smaller peptides as the hydrolysis time progressed. We hypothesize that a higher degree of hydrolysis due to the increased time of hydrolysis is often related to further enzyme activity on the reaction substrates. In contrast, the progression of hydrolysis-reaction from 2 h to 4 h (*p* > 0.05) of protease-generated hydrolysates did not show a very rapid increase in the displayed DH. However, after progressing the bio-catalysis reaction to 6 h, a significantly higher DH (*p* < 0.05) was recorded, as presented in Table 1. A similar hydrolysis reaction where increasing hydrolysis up to 4 h was found to be slower than between 4 to 6 h of hydrolysis [6].

The level of DH values obtained in this study remains comparable to those of other studies. For instance, the hydrolysis of quinoa proteins after 2 h of hydrolysis by pepsin pancreatin and papain was noticed to be in the range of 15–35% [7]. Similarly, 4 h of hydrolysis by Alcalase reported a DH value of 48% [14]. The variations noticed between DH produced by different enzymes after a similar time of hydrolysis could be ascribed to the specificity and substrate affinity of each enzyme towards the quinoa protein substrate [16].

### 3.2. Cholesterol Esterase Inhibitory Activity

QPHs were analyzed for their ability to inhibit CEase. CEase is a known polymeric enzyme existing in the bile that initiates dietary cholesterol esters’ hydrolytic reaction, thereby releasing cholesterol and free fatty acids [28]. Inhibiting CEase can indirectly prevent cholesterol absorption by the human body by diminishing cholesterol’s release from dietary lipids. The CEase inhibitory potential of generated QPHs and QPI were measured in terms of IC_50_ (half maximal inhibitory concentrations), and the results are illustrated in Table 1. The low IC_50_ value of hydrolysates connotes high CEase inhibitory activity. Upon hydrolysis, the QPHs derived from bromelain showed an increased CEase inhibition, with progression of the bio-catalysis reaction from 2 to 6 h (IC_50_ values = 0.67 mg/mL to 0.64 mg/mL to 0.63 mg/mL, respectively). Therefore, bromelain-generated hydrolysates with high CEase inhibitory activity can be developed at longer hydrolysis times. On the other hand, the CEase inhibitory potential of chymotrypsin and protease derived hydrolysates decreased with the increased bio-catalysis reaction time from 2 to 6 h.

From this study, it was evident that chymotrypsin-generated QPHs showed the highest CEase inhibitory activity with IC_50_ values of 0.51 mg/mL, 0.55 mg/mL, and 0.55 mg/mL after 2, 4, and 6 h of hydrolysis, respectively. However, there was no significant difference (*p* > 0.05) observed between hydrolysates generated at 4 h and 6 h hydrolysis time. Similar results were obtained by [6], where chymotrypsin exhibited higher activity compared to other enzymes used, and this implies that it is more effective in hydrolyzing quinoa protein. This was attributed to the broad specificity of chymotrypsin enzymes to generate small-sized peptide and free amino acids. In addition, the variation exhibited in the IC_50_ values of hydrolysates with respect to different time periods could be explained with the ability of specific enzymes to either degrade or generate CEase inhibitory peptides during hydrolysis [29]. The protease-generated hydrolysates showed a lower CEase inhibitory activity in comparison to bromelain and chymotrypsin generated hydrolysates. A possible explanation could be that the cleavage sites that the enzyme binds to was not fully exposed, which consequently reduced the inhibition effect. However, the protease hydrolyzed under hydrostatic pressure processing showed an increased inhibition rate of up to 49.1% [30].

Unhydrolyzed quinoa protein, i.e., intact quinoa protein (QPI), showed the least CEase inhibitory activity (IC_50_ value of 1.01 mg/mL). A study of the CEase-inhibitory activity of amaranth protein recently reported in our lab also showed a significant low CEase-inhibitory effect of amaranth protein isolate in comparison to IC_50_ values recorded for hydrolysates [13]. The hydrolysate with the utmost CEase inhibitory IC_50_ value of 0.51 mg/mL (chymotrypsin-derived for 2 h) was approximately two times lower than the IC_50_ value (1.01 mg/mL) of QPI, indicating that derived hydrolysates can therefore inhibit CEase two-fold more than their intact protein counterpart. Furthermore, the results suggested an overall high CEase inhibitory activity of QPHs compared to intact quinoa protein, which is evident from the lower IC_50_ values observed. This is similar to the study of Jafar, et al. [31] that reported the higher CEase inhibitory activity of protein hydrolysates derived from camel whey compared to intact protein from Camel whey.

### 3.3. Pancreatic Lipase (PL) Inhibitory Activity

Pancreatic lipase is the primary enzyme that contributes largely to the digestion of dietary lipids, and the only effective way to adjust lipid absorption is by inhibiting the enzyme [32]. The pancreatic lipase inhibitory activity IC_50_ values for QPHs (bromelain, chymotrypsin, protease) and un-hydrolyzed quinoa protein are depicted in Table 1. Results from this study showed that the PL-inhibitory IC_50_ values varied from 0.90 to 10.4 mg/mL, implying a substantial ability to inhibit PL activity. The PL-inhibitory activity of all the hydrolysates (0.90–3.32 mg/mL) exhibited significantly higher inhibition than the QPI, indicating that hydrolysis efficiently increased the inhibitory activity of quinoa protein. Hydrolysate QC-6 demonstrated the highest PL-inhibitory activity (0.78 mg/mL), followed by QB-6, recording IC_50_ values of 0.90 mg/mL, respectively. All chymotrypsin and bromelain generated hydrolysates showed higher PL-inhibitory activities than the protease generated hydrolysates. The different inhibitory activity observed among hydrolysates generated from different enzymes (bromelain, chymotrypsin and proteases) could be explained with the specificity and varying degree of hydrolysis among other factors of the individual enzyme [5].

On the other hand, a significant effect (*p* < 0.05) was observed in the pancreatic lipase inhibitory activity of QPHs upon hydrolysis with different enzymes at different hydrolysis times. A significant increase in the PL inhibitory activity of QPHs was observed with the increase in hydrolysis time progression, which is shown with lower IC_50_ values. For instance, chymotrypsin derived hydrolysate with the most potent PL-inhibitory activity IC_50_ values decreased from 2.90 to 0.78 mg/mL as the hydrolysis time increased. In the past, various studies have demonstrated an increase in the enzyme inhibitory potential of protein hydrolysates with the increase in the duration of hydrolysis. This implies that the progressive time of hydrolysis would sufficiently release peptides that might bind the active site of pancreatic lipase, thus inhibiting lipase enzymes. In addition, hydrolysates generated from chymotrypsin showed the maximum inhibitory activity in comparison to bromelain and proteases. This result upholds the potent characteristic of chymotrypsin generated hydrolysates, as reported by [6,16].

Furthermore, overall, QPHs showed the highest pancreatic lipase inhibitory activity in comparison to unhydrolyzed samples, and this accounted for an approximately 110% reduction in the IC_50_ value. This result agrees with the findings of Mudgil, et al. [33], who reported an increased PL-inhibitory activity of milk proteins after a bio-catalysis reaction using bromelain, chymotrypsin, alcalase and papain. Similarly, [34] had previously shown the potential of quinoa as a coadjutant therapeutic agent of cardiovascular diseases. The authors also reported quinoa’s ability to minimize lipid profile and glucose levels that are initiated by fructose which often leads to most of the well-known detrimental effects in humans. The use of derived hydrolysates from quinoa seeds comprising grains greater than 10% was reported to significantly lessen high levels of plasma and liver total cholesterol in nourished mice [35]. Consequently, QPHs can substitute synthetic drugs as anti-hyperlipidemia therapeutic agents, having shown excellent potential towards PL-inhibition.

### 3.4. Selection and Identification of Cholesterol Esterase Inhibitory Peptides from Selected QPHs

Based on the overall higher CEase and PL inhibitory activities of chymotrypsin generated hydrolysates obtained at the highest hydrolysis time of 6 h (QC-6), it was further chosen for peptide identification by LC-MS Q T-O-F. Additionally, the ABTS radical scavenging and anti-hemolytic activities shown by this specific hydrolysate reported in a previously published work by [6] made it an exciting hydrolysate to characterize its existing bioactive peptides. The identified peptides were classified as bioactive peptides by their scoring of >0.80 on a web server known as Peptide Ranker and further exploration into their in silico interaction with target enzymes using the Pepsite 2 web server [19].

In total, 136 peptides were identified (Appendix A), however, only 35 peptides were shortlisted to be biologically active based on a Peptide Ranker score greater than 0.8 (Table 2), and these peptides were subjected to an in-silico mode for further structure-activity analysis. The potential active binding sites of the identified peptides on the binding sites of CEase enzyme were accessed based on a statistical significance of 5% (Table 2). The significant variation (*p* < 0.05) implies that peptides derived from QC-6 significantly bound to the hotspot sites of CEase, resulting in non-competitive inhibition due to the modification in protein interaction networks or loss of enzyme activity that blocked its catalytic binding sites and substrate [19]. The results obtained from PepSite 2 showed that the peptides FFE, DFTF, DFLM, ML, CDCP, CYTF, QHPHGLGALCAAPPST, LR, RR, HVQGHPALPGVPAHW, AGLR, FTVM, LLPYH, ASNLDNPSPEGTVM, and HMLH had a significant (*p* < 0.05) binding effect on CEase (Table 2). However, QHPHGLGALCAAPPST, HVQGHPALPGVPAHW, and ASNLDNPSPEGTVM among the listed peptides were observed to be capable of binding the active sites of CEase. The most potent identified peptides HVQGHPALPGVPAHW and QHPHGLGALCAAPPST that could bind up to 13 and 14 bound residues were selected for mass spectrometry analysis. Two to sixteen amino acid residues were observed in the thirty-five peptides recorded (Table 2). This number is within the range of anti-hypercholesterolemic peptides reported by [36] for β-lactoglobulin peptides. Previous studies have suggested that peptides that can bind to more than eight residues have the ability to effectively inhibit the activity of antidiabetic and antihypertensive properties [37]. This connotes that the identified peptides (QHPHGLGALCAAPPST, HVQGHPALPGVPAHW, ASNLDNPSPEGTVM) from this study have the potential of inhibiting the activity of CEase.

According to [38], the active binding sites of CEase, which is vital for its catalytic function, contain an oxyanion hole (glycine and alanine), the catalytic triad (histidine and serine), and esteratic site. Some generated bioactive peptides from QC-6 derived hydrolysates could bind to part of the active site containing the catalytic triad, specifically His435 and Ser194. For instance, QHPHGLGALCAAPPST, HVQGHPALPGVPAHW, ASNLDNPSPEGTVM, HVASGAGPW, and KPGGTAGSALPRPAHW could bind up to fourteen, thirteen, thirteen, nine, and six bound residues of CEase, respectively. This result is consistent with previous studies where peptides that have shown higher inhibition potential to His435 and Phe324 residues are categorized as CEase inhibitors because they play a major role in the binding of cholesterol [29]. Similarly, [32], stated that peptides derived from Camel milk whey hydrolysates that could bind to residues His435 and Phe325 are important CEase inhibitors. Likewise, peptide HVASGAGPW could only bind to residue Phe324, which also suggests a potential CEase inhibitor.

Furthermore, [29] had earlier predicted the binding site residues of nine inhibitors of CEase using molecular docking to human and bovine CEase. The observation from this peptide sequencing indicated that the identified peptide sequence was able to form a hydrogen bonding with Gly107 in human CEase and formed a hydrophobic interaction with Ala108 in both human and bovine CEase. In addition, the Trp227 and Phe 324 of bovine CEase was observed to form cation-π interactions and π-stacking with the Phe1 residue of the peptide. Similarly, here Ala108, Phe324, and Trp227 were the major residues with active hotspots, except for Gly107 residues with no active site. This suggests that the most potent peptides QHPHGLGALCAAPPST, HVQGHPALPGVPAHW and ASNLDNPSPEGTVM could bind Ala108, Ser194, Ala195, Trp227, Phe324 and His435 in the active site of CEase, which could suggest an excellent bioactive peptide with prospects of inhibiting CEase, thus limiting dietary cholesterol absorption, which is anticipated for a CEase inhibitor.

Moreover, the CEase inhibitory capacity can further be explained with existing hydrophobic amino acids, and residue near the C-terminal and N-terminal end in the sequenced peptides. A previous study reported the major role the hydrophobicity of peptides plays in the anti-hypercholesterolemic activity of peptides, which is mainly the binding of bile acids. Identified peptides with anti-hypercholesterolemic ability inhibit bile acids absorption in the ileum; as a result, they reduce the level of blood cholesterol concentration [30]. Hydrophobic amino acids, namely alanine, leucine, phenylalanine, proline, etc. have been directly associated with the hypercholesterolemic-lowering activity of peptides. Similarly, peptides with specific hydrophobic amino acids at their terminal ends have demonstrated antidiabetic inhibitory potential. In the present study, aliphatic (Leu and Ala) and aromatic (Phe, Tyr and Trp) hydrophobic amino acids were identified, which are desirable at the C-terminal of the peptide. A previous study identified peptide FDGEVK derived from β-lactoglobulin tryptic hydrolysate as a potential anti-hypercholesterolemic peptide due to the presence of two such C-terminal amino acids [36]. Therefore, bioactive peptides derived from quinoa proteins could potentially inhibit CEase enzymes.

### 3.5. Identification of PL Inhibitory Peptides

Peptides with PL inhibitory potentials are shown in Table 3. From this study, results show that the peptides ML, MLLL, QHPHGLGALCAAPPST, LPLLR, MFVPVPH, HVQGHPALPGVPAHW, FTVM, LLPYH, MVLP, and HMLH derived from QC-6 showed a significant (*p* < 0.05) binding effect to PL. In this study, the number of peptides derived from QC-6 was 35, and the number of amino acids ranged from 2 to 16. This value is comparable to the range reported by [33] for PL-inhibiting peptides derived from Camel milk protein hydrolysates and the range reported by [39] for pinto beans. However, this was lower than the amino acid value of twenty-three derived from Cumin seeds [40]. The results indicated two peptides (QHPHGLGALCAAPPST and HVQGHPALPGVPAHW) with the most significant (*p* < 0.05) potent PL-inhibitory potential, binding up to 14 and 13 important amino acids, suggesting superior linkages to the target enzyme (Table 3).

Furthermore, these peptides have also demonstrated CEase inhibitory potential, as explained above, implying that these peptides have valuable anti-hypercholesterolemic properties. Other peptides (FSAGGLP, MFVPVPH, and ASNLDNPSPEGTVM) have shown a similar potent inhibitory potential binding up to 11, 11, and 12 important amino acids of PL, respectively. These peptides could be considered as potential PL inhibitors due to their maximum interaction with the binding sites of PL enzyme. Similarly, some peptides showed a high number of binding sites, thus possessing promising PL-inhibitory activity. Peptide HMCH could interact with nine bound residues, four peptides (LPLLR, FTVM, LLPYH, and MVLP) with eight bound residues, and peptide DFLM interacting with six bound residues. Among all the peptides that significantly (*p* < 0.05) bound to the active sites, only peptide ML interacted with a lower number of hotspots (four sites), and can be regarded as a poor PL inhibitor.

Furthermore, all six of the most potent peptides that interacted with maximum bound residues could bind Phe78, Ser153, Phe216, His264, and His152 from the hotspot sites of PL. The other peptides (HMCH, LPLLR, FTVM, LLPYH, and MVLP, DFLM) were capable of binding to similar active sites. Similar binding spots for PL inhibitory activity from cumin seeds have been reported by [40], and camel whey protein hydrolysates by [29]. Moreover, the peptides DFLM, KIVLDSDDPLFGGF, LPLLR, FTVM, LLPYH, MVLP, ASNLDNPSPEGTVM, and HMLH showed the ability to bind to similar hotspots (Phe78, Ser153, His152, Phe216, and His264). Furthermore, peptides showed the potential to indirectly inhibit PL by binding Try115, Leu154, Gly155, Ala179, and Pro181, which do not belong to the characteristic catalytic triad of pancreatic lipase.

It was observed that peptides that have the potential to bind a higher number of active sites of lipase were the hydrophobic amino acids. A majority of the peptides that exhibited the potential of binding a high number of binding sites of lipase consisted of hydrophobic amino acid residues such as proline and leucine. This is comparable to a previous study that showed proline and leucine to be the most active residue bound of lipase inhibiting peptides derived from pinto beans [39]. In addition, hydrogen bonding can also influence hydrophilic amino acid to bind generated bioactive peptides inhibitors to PL [40]. The PL inhibitory method of derived peptides from seed proteins has been reportedly measured against the mechanism of a synthetic lipase inhibitor (Orlistat), and a similar PL inhibitory mechanism observed between them [39,40]. Thus, novel QPHs derived from chymotrypsin at 6 h have PL-inhibitory peptides with the potential to bind active sites of lipase enzyme. This study provides novel PL inhibitory peptdies derived from quinoa that could be health-promoting in combating hypercholesterolemia.

### 3.6. Molecular Docking of Shortlisted Peptides in the Active Site of Human and Bovine CEase

To elucidate the binding pose and to identify the underlying intermolecular interactions of potential quinoa derived peptides in the active site of human and bovine CEase, molecular docking was performed. Table 4 details the GlideScore, MM-GBSA binding free energy, and the observed intermolecular interactions of the top ranked peptides. It was observed that the peptide HVASGAGPW docked to human CEase with a GlideScore of −9.07 kcal/mol and an MM-GBSA binding free energy of −64.32 kcal/mol, while the peptides AHCGGLPY, LYNDWDLR, MFVPVPH and FSAGGLP exhibited GlideScores of −8.95, −8.9, 8.3 and 7.4 kcal/mol, respectively.

CEase is a member of the alpha/beta hydrolase fold family. In the human CEase protein, the active site region includes an oxyanion hole comprising Gly107-Ala108 and Ala195 residues, and a catalytic triad containing Ser194-His435-Asp320 residues; both regions are essential for the functional activity of the protein. The hydroxyl group of Ser194 functions as a nucleophile that facilitates the hydrolytic action [41]. Apart from this, two acidic amino acids, Asp434 and Glu437, are positioned close to the active site region which renders a negative charge to the whole catalytic domain of the protein.

Among the docked peptides, the top docked peptide HVASGAGPW fits perfectly in the active site and is stabilized by hydrogen and hydrophobic bonds. Interestingly, Gly107, Ala108, and Ala195 residues, of the oxyanion hole, and His435, from the catalytic triad, were involved in the strong binding of this peptide with human CEase. The peptide AHCGGLPY showed interactions with several residues surrounding the active site including Ala108. Similarly, other top-docked quinoa peptides such as LYNDWDLR, MFVPVPH and FSAGGLP have shown interactions with the residues involved in the active site (Figure 1). These interactions can further block the catalytic activity of human CEase, supporting their potential as human CEase inhibitors. Like human CEase, bovine CEase shares similar residues (Ser194-His435-Asp320) in the catalytic triad and oxyanion hole of the molecule [42]. Sequence analysis revealed that the residue identity of human CEase and bovine CEase is 80%, and the overall three-dimensional structure of the active site region closely resembles human CEase [43]. The docked peptide AHCGGLPY exhibited a GlideScore of −10.05 kcal/mol and an MM-GBSA-based binding free energy of −72.14 kcal/mol. The active site residues Ala108 and Ser194 formed hydrogen and hydrophobic interactions with this peptide. Following this, the peptides MFVPVPH, FSAGGLP, HVASGAGPW and LPLLR also exhibited good GlideScores and MM-GBSA values (Table 4; Figure 1). The top peptides identified were observed to interact with the critical residues involved in the active site of the enzyme, which directly contributes to the inhibitory activity of the peptides and ligand interaction diagrams, which are represented in Figure 2.

### 3.7. Molecular Docking of Identified Peptides in the Active Site of Human and Porcine PL

The binding pose and intermolecular interaction of quinoa peptides with PL was elucidated using molecular docking simulations. GlideScore and calculated binding energy, based on MM-GBSA, for the best-docked poses as well as intermolecular interactions observed in the docked protein-peptide complex are provided in Table 5. Similar to several esterases and lipases, the PL active site is positioned in the N-terminal domain, which has a catalytic triad (Ser152, Asp176, and His263), and access to this site is controlled by a surface loop. Movement of the lid induces conformational changes in the protein structure that eventually leads to oxyanion hole unmasking that is involved in the interfacial binding region [44].

The binding mode of the best-docked peptide, HVASGAGPW, within the active site of PL was analyzed. Results indicated that the peptide docked to the enzyme recorded a GlideScore of −11.74 kcal/mol and an MM-GBSA binding energy of −46.02 kcal/mol. The other top docked peptides AHCGGLPY, FSAGGLP, LLPYH and MFVPVPH also exhibited good GlideScores and binding energies (Table 5). It was observed that peptide AHCGGLPY recorded a GlideScore of −11.73 kcal/mol, and −61.96 kcal/mol for the calculated binding energy. FSAGGLP recorded −9.72 kcal/mol and −79.39 kcal/mol for the GlideScore and MM-GBSA binding energy, respectively. The GlideScores for peptides LLPYH and MFVPVPH were −9.72 kcal/mol and −8.82 kcal/mol, respectively, and the MM-GBSA binding energies were −77.18 kcal/mol and −70.12 kcal/mol, respectively.

All of the reported peptides exhibited interactions with critical amino acids in the catalytic binding pockets of the enzyme by forming several hydrogen and hydrophobic interactions (Figure 3). Similar to human PL, the active site of porcine PL comprises the catalytic domain, consisting of Ser153, Asp177, and His264. Similar to human PL, the peptides HVASGAGPW, AHCGGLPY, MFVPVPH and FSAGGLP exhibited good GlideScores and binding energies. Apart from these peptides, LPLLR and CYTF also docked well in the active site of the protein. It was observed that the peptide HVASGAGPW was well-docked in the catalytic region by exhibiting −10.19 kcal/mol and −80.44 kcal/mol for the GlideScore and binding energy, respectively. Followed by this, the peptides AHCGGLPY and MFVPVPH had GlideScores of −10.95 and −8.58 kcal/mol and a binding energy of −53.03 and −67.89 kcal/mol. The peptides were stabilized primarily by electrostatic and hydrophobic interactions, with residues in the active site of the enzyme (Figure 3 and Figure 4).

## 4. Conclusions

This study investigated the CEase- and PL-inhibitory potential of QPHs as a functional ingredient for producing nutraceuticals that can be used to manage hypercholesterolemia. This current study explored the generation of hydrolysates from quinoa proteins using three enzymatic approaches (chymotrypsin, proteases, and bromelain). The three enzymes explored produced hydrolysates with higher inhibitory activities towards CEase and PL compared to unhydrolyzed quinoa proteins. Nonetheless, chymotrypsin-generated hydrolysates at 6 h hydrolysis time exhibited higher CEase and PL inhibitory activity as confirmed by lower IC_50_ values. Findings showed that QPHs effectively inhibited the two enzymatic markers responsible for hypercholesterolemia by disrupting the enzyme-substrate interactions at the enzyme hotspot site. Novel peptide ASNLDNPSPEGTVM was found to have the best potential to be a CEase inhibitor, while FSAGGLP, KIVLDSDDPLFGGF, and MFVPVPH were found to be the potential PL inhibitors. The peptides QHPHGLGALCAAPPST and HVQGHPALPGVPAHW we observed to be potential inhibitors against both CEase and PL. In silico data showed that Gly107, Ala108, Ala195, and His435 were involved in the strong binding of these peptide with human CEase. Thus, peptides generated from quinoa seed protein may have a valuable application in ameliorating hypercholesterolemia based on predictions using software tools and molecular docking studies. However, further studies using in vitro, in vivo and cell line assays should be carried out to validate the potential anti-hypercholesterolemic activities observed in this work.

## Figures and Tables

**Figure 1 foods-12-01327-f001:**
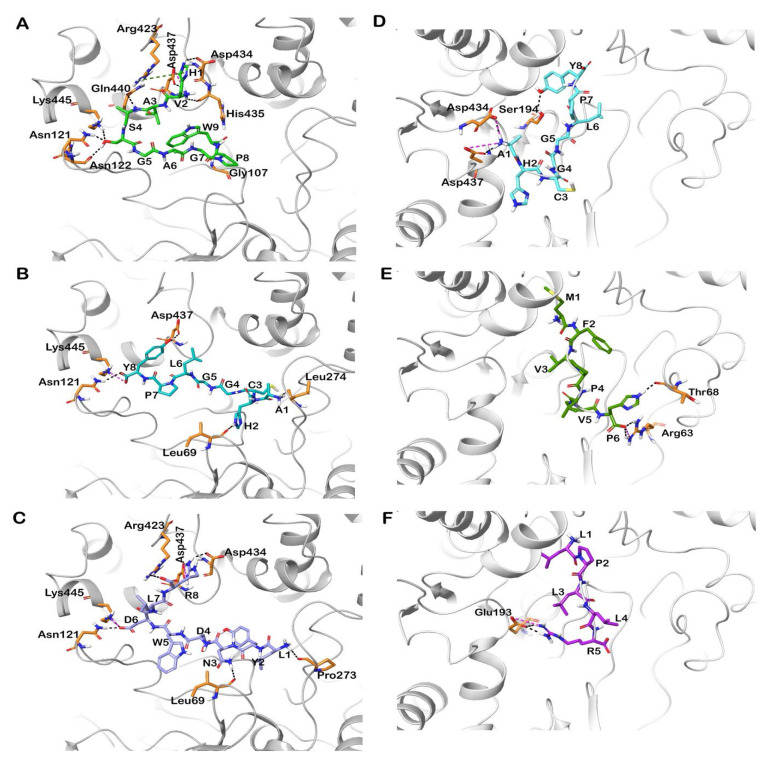
Docked poses of peptides in the active site of human and bovine CEase (PDB IDs: 1F6W and 1AQL). The protein is shown in cartoon representation and the docked peptides are shown in stick representation. The interactions of peptides (**A**) HVASGAGPW, (**B**) AHCGGLPY, (**C**) LYNDWDLR in the active site human CEase. Interactions of peptides (**D**) AHCGGLPY, (**E**) MFVPVPH and (**F**) LPLLR in the active site of bovine CEase. Hydrogen bonds, π-π stacking, π-cation interactions and salt bridges are represented in black blue, green and pink dashed lines. Keywords: Cholesterol esterase (CEase).

**Figure 2 foods-12-01327-f002:**
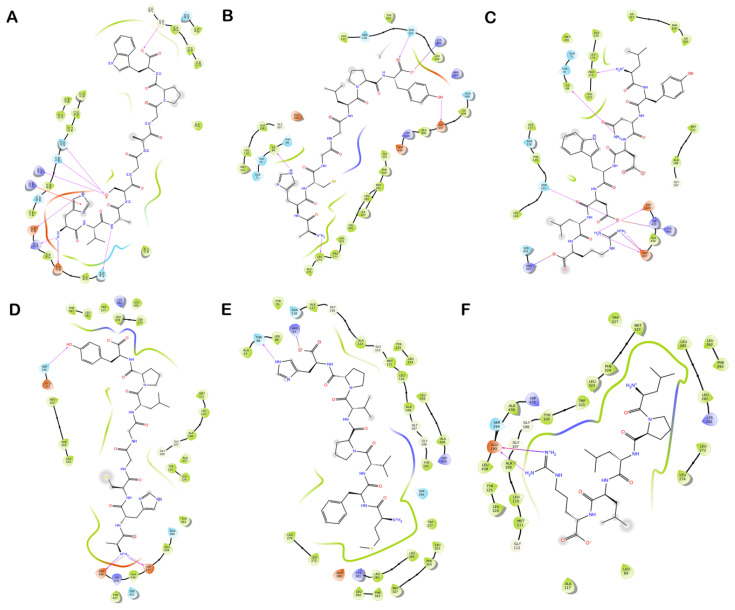
Ligand interaction diagram of inhibitory peptides with the residues of the enzyme CEase. The interactions of peptides (**A**) HVASGAGPW, (**B**) AHCGGLPY, and (**C**) LYNDWDLR in the active site human CEase. Interactions of peptides (**D**) AHCGGLPY, (**E**) MFVPVPH and (**F**) LPLLR in the active site of bovine CEase. Purple circles represent positively charged amino acids, red circles represent negatively charged amino acids, green circles represent hydrophobic amino acids, and light blue circles represent polar amino acids. Purple dashed arrows represent hydrogen bonds involving amino acid side chain and regular purple arrows represent hydrogen bonds involving the amino acid backbone. π-π interactions are represented with green lines.

**Figure 3 foods-12-01327-f003:**
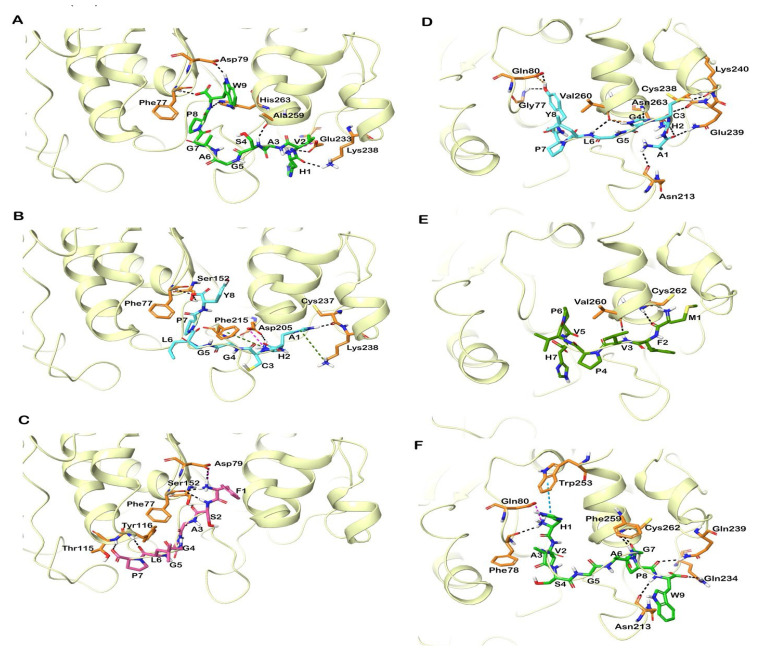
Docked poses of peptides in the active site of human and porcine PL (PDB ID: 1LPB, 1ETH). The protein is shown in cartoon representation and the docked peptides are shown in stick representation. The interactions of peptides (**A**) HVASGAGPW, (**B**) AHCGGLPY, and (**C**) FSAGGLP in the active site of human PL. Interactions of peptides (**D**) AHCGGLPY, (**E**) MFVPVPH and (**F**) HVASGAGPW in the active site of porcine PL. Hydrogen bonds, π-π stacking, π-cation interactions and salt bridges are represented in black-blue, green and pink dashed lines. Keyword: Pancreatic lipase (PL).

**Figure 4 foods-12-01327-f004:**
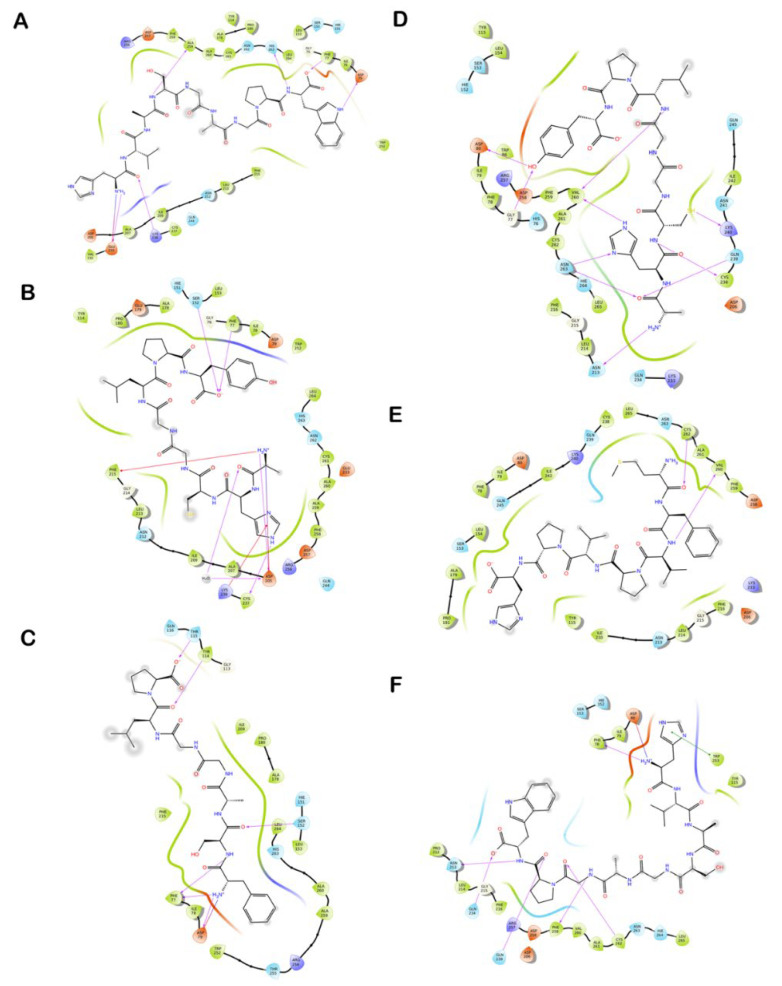
Ligand interaction diagram of inhibitory peptides with the residues of the enzyme PL. The interactions of peptides (**A**) HVASGAGPW, (**B**) AHCGGLPY, and (**C**) FSAGGLP in the active site of human PL. Interactions of peptides (**D**) AHCGGLPY, (**E**) MFVPVPH and (**F**) HVASGAGPW in the active site of porcine PL. Purple circles represent positively charged amino acids, red circles represent negatively charged amino acids, green circles represent hydrophobic amino acids, and light blue circles represent polar amino acids. Purple dashed arrows represent hydrogen bonds involving the amino acid side chain and regular purple arrows represent hydrogen bonds involving the amino acid backbone. π-π interactions are represented with green lines.

**Table 1 foods-12-01327-t001:** DH, CEase and PL IC_50_ inhibitory activities of quinoa protein hydrolysates generated by bromelain, chymotrypsin, and protease after hydrolysis reaction.

Sample	DH (%)	CEase (mg/mL)	PL (mg/mL)
QPI	-	1.01 ± 0.03 ^e^	10.4 ± 0.14 ^d^
QB-2	34.99 ± 0.92 ^a^	0.67 ± 0.03 ^c^	1.97 ± 0.03 ^b^
QB-4	41.91 ± 0.36 ^b^	0.64 ± 0.02 ^bc^	1.03 ± 0.01 ^a^
QB-6	51.91 ± 2.03 ^c^	0.63 ± 0.01 ^bc^	0.90 ± 0.00 ^a^
QC-2	36.01 ± 3.08 ^a^	0.51 ± 0.01 ^a^	2.90 ± 0.11 ^c^
QC-4	44.09 ± 0.47 ^b^	0.55 ± 0.00 ^ab^	1.51 ± 0.06 ^ab^
QC-6	66.05 ± 1.60 ^d^	0.55 ± 0.01 ^ab^	0.78 ± 0.04 ^a^
QP-2	45.05 ± 1.00 ^b^	0.80 ± 0.02 ^d^	3.32 ± 0.52 ^c^
QP-4	46.01 ± 0.80 ^b^	0.88 ± 0.02 ^d^	1.53 ± 0.03 ^ab^
QP-6	63.3 ± 2.66 ^d^	0.87 ± 0.01 ^d^	1.08 ± 0.02 ^ab^

Keynote: QPI: Intact quinoa protein isolate; QB-2, QC-2 and QP-2 denotes bromelain, chymotrypsin, and protease 2 h generated hydrolysate; QB-4, QC-4 and QP-4 denotes bromelain, chymotrypsin and protease 4 h generated hydrolysate; QB-6, QC-6 and QP-6 denotes bromelain, chymotrypsin and protease 6 h generated hydrolysate, respectively. DH: Degree of hydrolysis; CE: Cholesterol esterase; and PL: Pancreatic lipase. Different small letters in the column indicates significant different amongst unhydrolysed and hydrolysed QPHs.

**Table 2 foods-12-01327-t002:** Pepsite 2 analysis of the selected peptides using cholesterol esterase as a protein receptor.

Peptide	*p*-Value	Peptide Residues	Bound Residues
FFE	0.04342	F1, F2, E3	Tyr7, Phe12
DFTF	0.03113	D1, F2, T3, F4	Phe235, Trp236, Tyr279, His283, Tyr284
DFLM	0.01757	D1, F2, L3, M4	Tyr7, Phe12
ML	0.04441	M1, L2	Phe235, Trp236, His283
CDCP	0.04734	C1, C3, P4	Tyr7, Phe12
CYTF	0.01539	C1, Y2, T3, F4	Tyr7, Phe12
FSAGGLP	0.3886	G5, L6, P7	Phe235, Trp236, His283, Tyr284
MLLL	0.05859	M1, L2, L3, L4	Tyr7, Phe12
MYLY	0.1116	M1, Y2, L3, Y4	Tyr7, Phe12
QHPHGLGALCAAPPST	0.04033	Q1, H2, P3, G7, L9, C10, P14, S15	Gy107, Ala108 *, Ser194 *, Ala195 *, Ser220, Trp227 *, Phe235, Trp236, His283, Tyr284, Leu285, Phe324 *, Phe393, His435 *
KIVLDSDDPLFGGF	0.2461	I2, V3, D7, D8, P9, L10, F11	Ala108 *, Ser194 *, Trp227 *, Phe235, Trp236, His283, Tyr284, Leu285, Phe324 *, Phe393, His435 *
AHCGGLPY	0.2596	A1, H2, C3, G4	Tyr7, Phe12
LYNDWDLR	0.379	W5, D6, L7, R8	Tyr7, Phe12
PGGAR	0.2249	G3, A4, R5	Phe235, Trp236, His283, Tyr284
SGPAR	0.08677	P3, A4, R5	Tyr7, Phe12
LPLLR	0.04626	P2, L3, L4, R5	Tyr7, Phe12
KPGGTAGSALPRPAHW	0.3388	K1, P2, G3, L10	Phe235, Tyr236, Tyr279, His283, Tyr284, Phe324 *
FE	0.06821	F1, E2	Tyr7, Phe12
HDSF	0.09456	H1, D2, S3, F4	Tyr7, Phe12
LR	0.04441	L1, R2	Tyr7, Phe12
VYML	0.08002	V1, Y2, M3, L4	Tyr7, Phe12
RR	0.02794	R1, R2	Tyr7, Phe12
LRL	0.1121	L1, R2, L3	Tyr7, Phe12
MFVPVPH	0.3249	P4, V5, P6	Phe235, Trp236, His283, Tyr284
HVQGHPALPGVPAHW	0.02606	H1, V2, P6, A7, L8, P9, G10, P12, A13, H14	Ala108 *, Ser194 *, Ala195 *, Ser220, Trp227 *, Phe235, Trp236, His283, Tyr284, Leu285, Phe324 *, Phe393, His435 *
AGLR	0.02471	A1, G2, L3, R4	Tyr7, Phe12
HVASGAGPW	0.3417	V2, A3, P8, W9	Ala108 *, Ser194 *, Ala195 *, Trp227 *, Phe235, Trp236, His283, Tyr284, Phe324 *
FTVM	0.02389	F1, T2, V3, M4	Phe235, Trp236, His283, Tyr284
LLPYH	0.04167	L2, P3, Y4, H5	Tyr7, Phe12
MVLP	0.1188	V2, L3, P4	Phe235, Trp236, His283
GARR	0.1087	G1, A2, R3	Phe235, Trp236, His283, Tyr284
ASNLDNPSPEGTVM	0.03203	D5, N6, P7, S8, E10, T12, V13, M14	Ala108 *, Ser194 *, Trp227 *, Phe235, Trp236, Arg239, Tyr279, His283, Tyr284, Leu285, Phe324 *, Phe393, His435 *
CVLSPL	0.2097	V2, S4, P5, L6	Phe235, Trp236, Tyr279, His283, Tyr284, Val353
HMLH	0.02838	H1, M2, L3, H4	Tyr7, Phe12
CALVGL	0.2557	C1, A2, V4	Tyr7, Phe12

* Hotspots that could inhibit the activity of cholesterol esterase if bounded by the peptide.

**Table 3 foods-12-01327-t003:** Pepsite 2 analysis of the selected peptides using pancreatic lipase as a protein receptor.

Peptide	*p*-Value	Peptide Residues	Bound Residues
FFE	0.06672	F1, F2, E3	Phe78 *, Tyr115, Ser153 *, Pro181, Phe216 *
DFTF	0.1717	F2, T3, F4	Phe78 *, Tyr115, Ser153 *, Pro181, Phe216 *
DFLM	0.08332	F2, L3, M4	Phe78 *, Tyr115, Ser153 *, Pro181, Phe216 *, His264 *
ML	0.008899	M1, L2	Phe78 *, Tyr115, Ser153 *, Phe216 *
CDCP	0.2673	C1, D2, C3	Lys81, Glu84, Trp253
CYTF	0.1707	C1, Y2, T3, F4	Phe78 *, Tyr115, Pro181, Ile210, Phe216 *
FSAGGLP	0.3038	F1, S2, A3, G4	Gly77, Phe78 *, Tyr115, His152 *, Ser153 *, Leu154, Gly155, Ala179, Pro181, Phe216 *, His264 *
MLLL	0.009667	M1, L2, L3, L4	Gly77, Phe78 *, Tyr115, Ser153 *
MYLY	0.05041	M1, Y2, L3, Y4	Phe78 *, Tyr115, Ser153 *, Leu154, Ala179, Pro181, Ile210, Phe216 *, His264 *
QHPHGLGALCAAPPST	0.01253	Q1, H2, P3, H4, L9, C10, A11, A12, P13, P14, S15	Gly77, Phe78 *, Lys81, Tyr115, His152 *, Ser153 *, Leu154, Gly155, Ala179, Pro181, Phe216 *, His264 *, Trp253
KIVLDSDDPLFGGF	0.05873	K1, I2, V3, D5, S6, D7, P9, L10, F11, G13, F14	Gly77, Phe78 *, Ile79, Asp80, Lys81, Glu84, Tyr115, Ser153 *, Leu154, Pro181, Phe216 *, Trp253, Arg257, His264 *
AHCGGLPY	0.2518	G5, L6, P7, Y8	Gly77, Phe78 *, Tyr115, Ser153 *, Leu154, Pro181, Ile210, Phe216 *, His264 *
LYNDWDLR	0.5264	L1, Y2, N3, W5	Gly77, Phe78 *, Tyr115, His152 *, Ser153 *, Leu154, Gly155, Ala179, Pro181, Ile210, Phe216 *, His264 *
PGGAR	0.2735	P1, G2, A4	Gly77, Phe78 *, Tyr115, Ser153 *, Leu154, Ala179, Glu180, Pro181, Phe216 *, His264 *
SGPAR	0.1096	G2, P3, A4	Phe78 *, Tyr115, Ser153 *, Pro181, Phe216 *, His264 *
LPLLR	0.01133	L1, P2, L3, L4	Phe78 *, Tyr115, Ser153 *, Ala179, Glu180, Pro181, Phe216 *, His264 *
KPGGTAGSALPRPAHW	0.1243	K1, P2, G3, A9, L10	Phe78 *, Lys81, Glu84, Tyr115, Ser153 *, Phe216 *, Trp253, His264 *
FE	0.1965	F1, E2	Phe78 *, Tyr115, Ser153 *, Phe216 *
HDSF	0.2314	H1, S3, F4	Phe78 *, Tyr115, Ser153 *, Leu154, Ala179, Phe216 *, His264 *
LR	0.2794	L1, R2	Lys81, Glu84, Trp253
VYML	0.1071	V1, Y2, M3, L4	Phe78 *, Tyr115, Ser153 *, Leu154, Ala179, Pro181, Ile210, Phe216 *, His264 *
RR	0.2074	R1, R2	Lys81, Glu84, Trp253
LRL	0.4811	L1, L3	Phe78 *, Tyr115, Ser153 *, Ala179, Glu180, Pro181, Phe216 *, His264 *
MFVPVPH	0.01305	F2, V3, P4, P6, H7	Gly77, Phe78 *, Tyr115, His152 *, Ser153 *, Leu154, Gly155, Ala179, Pro181, Phe216 *, His264 *
HVQGHPALPGVPAHW	0.02747	H5, P6, A7, L8, P9, H16	Gly77, Phe78 *, Lys81, Tyr115, His152 *, Ser153 *, Leu154, Gly155, Ala179, Pro181, Phe216 *, Trp253, His264 *
AGLR	0.2014		Gly77, Phe78 *, Tyr115, Ser153 *, Leu154, Pro181, Phe216 *, His264 *
HVASGAGPW	0.1145	H1, A6, G7, P8, W9	Gly77, Phe78 *, Lys81, Glu84, His152 *, Ser153 *, Leu154, Gly155, Ala179, Phe216 *, Trp253, His264 *
FTVM	0.02278	F1, T2, V3, M4	Phe78 *, Try115, Ser153 *, Leu154, Ala179, Pro181, Phe216 *, His264 *
LLPYH	0.008942	L1, L2, P3, H5	Gly77, Phe78 *, Tyr115, Ser153 *, Leu154, Pro181, Phe216 *, His264 *
MVLP	0.006331	M1, V2, L3, P4	Phe78 *, Tyr115, Ser153 *, Ala179, Glu180, Pro181, Phe216 *, His264 *
GARR	0.1135	G1, A2, R3, R4	Lys81, Glu84, Trp253
ASNLDNPSPEGTVM	0.08445	A1, S2, N3, L4, N6, P7, S8, P9, E10, V13, M14	Phe78 *, Asp80, Lys81, Glu84, Tyr115, Ser153 *, Leu154, Ala179, Phe216 *, Trp253, Arg257, His264 *
CVLSPL	0.1236	L3, S4, P5, L6	Phe78 *, Tyr115, Ser153 *, Ala179, Glu180, Pro181, Phe216 *, His264 *
HMLH	0.008295	H1, M2, L3, H4	Gly77, Phe78 *, Tyr115, His152 *, Ser153 *, Leu154, Gly155, Ala179, Phe216 *
CALVGL	0.2596	A2, L3, V4, L6	Phe78 *, Tyr115, Ser153 *, Pro181, Ile210, Phe216 *, His264 *

* Hotspots that could inhibit the activity of lipase if bounded by the peptide.

**Table 4 foods-12-01327-t004:** Docking score (GlideScore), MM-GBSA binding free energy, and interacting residues of human and bovine CEase with the identified peptides.

Peptide	GlideScore (kcal/mol)	MM-GBSA Based Binding Free Energy (kcal/mol)	Hydrogen Bonds	Hydrophobic Interactions	Salt-Bridge	π-π Interactions	Cation-π Interactions
CEase (Human)—PDB ID: 1F6W							
HVASGAGPW	−9.07	−64.32	Gly107, Asn121, Asn122, Asp434, His435, Gln440, Lys445	Ile69, Ala108, Leu110, Met111, Ala117, Phe119, Tyr123, Leu124, Tyr125, Ala195, Ile323, Phe324, Ala436, Ile439, Tyr453	Asp437		Arg423
AHCGGLPY	−8.95	−55.00	Ile69, Asn121, Leu274, Asp437	Ile69, Ala108, Leu110, Met111, Ala117, Leu124, Val272, Pro273, Leu274, Ala275, Met281, Leu282, Val285, Ile323, Phe324, Ile327, Ala436, Ile439, Tyr453	Lys445		
LYNDWDLR	−8.90	−63.67	Ile69, Asn121, Pro273, Asp434, Asp437	Ile69, Ala108, Met111, Ala117, Phe119, Leu124, Val272, Pro273, Leu274, Ala275, Met281, Ile323, Phe324, Ile327, Ala436	Arg423, Asp437, Lys445		
MFVPVPH	−8.37	−94.06	Asn121, His435	Tyr105, Ala108, Met111, Ala117, Phe119, Leu124, Tyr125, Ala195, Trp227, Met281, Leu282, Val285, Ile323, Phe324, Ile327, Leu392, Phe393, Ala436, Ile439, Tyr453	Lys445		Trp227
FSAGGLP	−7.44	−47.70	His435, Asp437	Ala108, Leu110, Met111, Ala117, Leu124, Tyr125, Ala195, Trp227, Val272, Met281, Leu282, Tyr284, Val285, Phe324, Leu392, Phe393, Ala436, Ile439			
HDSF	−7.89	−44.43	Gly107, Ala108, Glu193, Asp437, Gln440	Ala108, Tyr125, Ala195, Trp227, Met281, Leu282, Val285, Ile323, Phe324, Leu392, Phe393, Ala436, Ile439, Tyr453	Asp437		
LRL	−7.81	−49.25	Glu193, Ser194	Trp103, Tyr105, Ala108, Leu110, Met111, Tyr125, Ala195, Trp227, Val272, Met281, Leu282, Tyr284, Val285, Phe324, Leu392, Phe393, Ala436, Ile439	Glu193		
LLPYH	−7.57	−47.61	Ile69, Glu193, His435, Asp437	Ile69, Ala108, Leu110, Met111, Leu124, Val272, Met281, Val285, Ile323, Phe324, Ile327, Ala436, Ile439	Asp437		
MYLY	−7.30	−46.79	Gly107, His435, Asp437	Ala108, Met111, Leu124, Ala195, Trp227, Met281, Leu282, Val285, Ile323, Phe324, Ile327, Leu392, Phe393, Ala436, Ile439	Asp437	Phe324	
CVLSPL	−7.64	−65.08	Gly107, Ser194, Ala436, Asp437	Ala108, Leu110, Ala117, Leu124, Ala195, Trp227, Val272, Met281, Leu282, Val285, Ile323, Phe324, Ile327Leu392, Phe393, Ala436, Ile439			
CEase (Bovine)—PDB ID: 1AQL							
AHCGGLPY	−10.05	−72.14	Ser194, Asp434, Asp437	Ala108, Leu110, Met111, Ala117, Leu124, Tyr125, Trp227, Leu272, Leu274, Leu285, Leu323, Phe324, Met327, Leu392, Phe393, Tyr427, Ala436, Leu439, Tyr453	Asp434, Asp437		
MFVPVPH	−9.828	−87.97	Thr68	Ala67, Leu69, Tyr75, Tyr105, Ala108, Leu110, Met111, Ala113, Ala117, Leu124, Tyr125, Trp227	Arg63		
				Leu272, Leu274, Leu282, Leu285, Leu323, Phe324, Met327, Leu392, Phe393, Ala436, Leu439			
LPLLR	−9.218	−78.64	Glu193	Leu69, Trp103, Tyr105, Ala108, Leu110, Met111, Ala117, Leu124, Tyr125, Trp227, Leu272, Leu274, Leu282, Leu285, Leu323, Phe324, Met327, Leu392, Phe393, Ala436, Leu439	Glu193		
CYTF	−8.261	−62.93	Glu193, Ser194, His435	Ala108, Leu110, Met111, Leu124, Tyr125, Ala195, Trp227, Leu272, Leu274, Leu282, Leu285, Leu323, Phe324, Met327, Leu392, Phe393, Ala436, Leu439	Glu193		
FSAGGLP	−8.861	−73.84	Ala117	Tyr75, Tyr105, Ala108, Leu110, Met111, Ala113, Ala117, Leu124, Tyr125, Ala195, Leu272, Leu274, Leu282, Leu285, Leu323, Phe324, Met327, Leu392, Phe393, Leu439		Trp227	
LLPYH	−8.437	−76.32	Gly106, Glu193	Ala108, Leu110, Met111, Tyr125, Ala195, Trp227, Leu272, Leu274, Leu282, Leu285, Leu323, Phe324, Met327, Leu392, Phe393, Ala436, Leu439		His435	Phe324, His435
CVLSPL	−8.980	−71.06	Ala113, Ala117, His435	Tyr75, Tyr105, Ala108, Leu110, Met111, Ala113, Ala117, Leu124, Tyr125, Ala195, Trp227, Leu272, Leu274, Leu282, Leu285, Leu323, Phe324, Met327, Leu392, Phe393, Ala436, Leu439			Trp227, Phe324
PGGAR	−8.680	−72.64	Glu193	Ala108, Leu110, Met111, Tyr125, Ala195, Trp227, Leu272, Leu274, Leu282, Leu285	Glu193		Trp227
				Leu323, Phe324, Met327, Leu392, Phe393, Ala436, Leu439			
HDSF	−8.348	−33.84	Ala108, Ser194, Ala436	Ala108, Leu124, Ala195, Trp227, Leu272, Leu282, Leu285, Leu323, Phe324, Leu392, Phe393, Tyr427, Ala436, Leu439	Asp434, His435		
HVASGAGPW	−8.884	−67.55	Arg63, Gly106, Asn118, Asn122, Asp434, His435, Ala436	Tyr75, Trp103, Met111, Ala113, Ala117, Phe119, Tyr123, Leu124, Tyr125, Leu323, Met424, Tyr427, Ala436, Leu439, Tyr453	Asp434, Asp437		

**Table 5 foods-12-01327-t005:** Docking score (GlideScore), MM-GBSA binding free energy, and interacting residues of human and porcine PL with the identified peptides.

Peptide	GlideScore (kcal/mol)	MM-GBSA Based Binding Free Energy (kcal/mol)	Hydrogen Bonds	Hydrophobic Interactions	Salt Bridge	π-π Interactions	Cation-π Interactions
LIP (Human)—PDB ID: 1LPB							
HVASGAGPW	−11.74	−46.02	Phe77, Asp79, Glu233, Lys238, Ala259, His263	Phe77, Ile78, Tyr114, Leu153, Ala178, Pro180, Ala207, Ile209, Leu213, Phe215 Val232, Cys237, Trp252, Phe258, Ala259, Ala260, Cys261, Leu264	Glu233		
AHCGGLPY	−11.73	−61.96	Phe77, Ser152, Asp205, Cys237	Phe77, Ile78, Tyr114, Leu153, Ala178, Pro180, Ala207, Ile209, Leu213, Phe215, Trp252, Phe258, Ala259, Ala260, Cys261, Leu264	Asp205		Phe215, Lys238
FSAGGLP	−9.726	−79.39	Phe77, Asp79, Tyr114, Thr115, Ser152	Phe77, Ile78, Tyr114, Leu153, Ala178, Pro180, Ile209, Phe215, Trp252, Ala259, Ala260, Leu264	Asp79		
LLPYH	−9.716	−77.18	Asp79, Tyr114	Phe77, Ile78, Tyr114, Leu153, Ala178, Pro180, Ile209, Leu213, Phe215, Trp252, Ala259, Ala260, Leu264, Tyr267	Asp79	Tyr114, Phe215	
MFVPVPH	−8.824	−70.12	Phe77, Lys238, Ala259, Asp205	Phe77, Ile78, Tyr114, Leu153, Ala178, Pro180, Ala207, Leu213, Phe215, Val232, Trp252, Phe258, Ala259, Ala260, Cys261, Leu264	Asp79	Phe215, His263	
HDSF	−8.305	−45.69	Asp79, Phe77	Phe77, Ile78, Tyr114, Leu153, Ala178, Pro180, Ile209, Leu213, Phe215, Trp252, Ala259, Ala260, Leu264	Asp79	Phe215	
CVLSPL	−8.170	−68.48	Phe77	Phe77, Ile78, Tyr114, Leu153, Ala178, Pro180, Ile209, Leu213, Phe215, Ala259, Ala260, Leu264	Asp79		
CYTF	−8.353	−74.89	Phe77, Asp79	Phe77, Ile78, Tyr114, Ala178, Pro180, Ile209, Val210, Leu213, Phe215, Ala259, Ala260, Leu264, Tyr267	Asp79		
DFTF	−7.716	−55.27	Phe77, Asp79, Ser152	Phe77, Ile78, Trp85, Tyr114, Leu153, Ala178, Pro180, Ile209, Phe215, Trp252, Ala259, Ala260, Leu264, Tyr267	Asp79		
MYLY	−8.085	−95.47	Phe77, Asp79	Phe77, Ile78, Tyr114, Leu153, Ala178, Pro180, Ile209, Val210, Leu213, Phe215, Trp252, Ala259, Ala260, Leu264	Asp79		
LIP (Porcine)—PDB ID: 1ETH							
AHCGGLPY	−10.95	−53.03	Gly77, Asp80, Asn213, Cys238	Phe78, Ile79, Trp86, Tyr115, Leu154, Leu214, Phe216, Cys238, Ile242, Phe259			
			Gln239, Lys240, Val260, Asn263	Val260, Ala261, Cys262, Leu265			
MFVPVPH	−8.583	−67.89	Val260, Cys262	Phe78, Ile79, Tyr115, Leu154, Ala179, Pro181, Ile210, Leu214, Phe216, Cys238, Ile242, Phe259, Val260, Ala261, Cys262, Leu265			
HVASGAGPW	−10.19	−80.44	Phe78, Asn213, Gln234, Gln239 Phe259, Cys262	Phe78, Ile79, Tyr115, Pro212, Leu214, Phe216, Trp253, Phe259, Val260, Ala261, Cys262, Leu265	Asp80	Trp253	
LPLLR	−8.602	−77.53	Phe78, Asp206	Phe78, Ile79, Tyr115, Ile210, Leu214, Phe216, Trp253, Phe259, Val260, Ala261, Cys262, Leu265	Asp80, Asp206		Phe216
CYTF	−8.164	−75.32	Phe78, Asp80	Phe78, Ile79, Tyr115, Leu154, Ala179, Pro181, Ile210, Phe216, Val260, Ala261, Leu265	Asp80	Phe78, Tyr115	
FSAGGLP	−8.144	−72.31	Asp206, Leu214, Phe259, Val260	Phe78, Tyr115, Leu154, Ala179, Pro181, Ile210, Leu214, Phe216, Cys238, Phe259, Val260, Ala261, Cys262	Asp206		
FFE	−8.202	−58.14	Phe78, Asp80	Phe78, Ile79, Tyr115, Leu154, Ala179, Pro181, Phe216, Trp253, Val260, Ala261, Leu265	Asp80	Phe216, Trp253	
CDCP	−8.166	−57.75	Phe78, Asp80	Phe78, Ile79, Tyr115, Ala179, Pro181, Ile210, Phe216, Trp253, Val260, Ala261, Leu265	Asp80		
HDSF	−7.751	−58.29	Phe78, Asp80	Phe78, Ile79, Tyr115, Leu154, Ala179, Pro181, Ile210, Phe216, Trp253, Val260, Ala261, Leu265	Asp80		
MYLY	−8.308	−76.41	Phe78, Asp80, Phe259	Phe78, Ile79, Tyr115, Leu154, Ala179, Pro181, Leu214, Phe216, Trp253, Phe259, Val260, Ala261, Leu265	Asp80	Phe216, His264	

## Data Availability

The data presented in this study are available in the supporting materials.

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
