# Peer review of "Novel Plant-Protein (Quinoa) Derived Bioactive Peptides with Potential Anti-Hypercholesterolemic Activities: Identification, Characterization and Molecular Docking of Bioactive Peptides"

_foods, 2023, doi:10.3390/foods12061327_

Round 1

Reviewer 1 Report

In order to generate novel bioactive peptides from quinoa protein hydrolysates, the authors tried three enzymatic hydrolysis methods. LC-MS QTOF and molecular docking were then used to identify the most potent inhibitors for cholesterol esterase and pancreatic lipase. Several novel peptide inhibitors were suggested. The manuscript is well-structured and the results are well presented.

(1) In line 209 on page 5, please provide the references after “... in the ligand-binding were identified from the literature.”

(2) Is it possible to calculate the uncertainty for the MM-GBSA binding energy?

Reviewer 2 Report

The paper is interesting

First of all the title must be corrected,” hypercholestermic”; moreover if the authors use the word identification , the authors must show the identified peptides

Results: the authors should include a SDSPAGE pattern of the quinoa hydrolysates

Where is the table with all the identified peptides sequences, which are then used in Peptide ranker? The authors must include this table in the supplementary materials. It is important to know what are all the peptides present in the sample QC6.

From paragraph 3.4 , all the discussion is related only to silico analysis; the authors should reduce paragraph 3.4 and the following 3.3 (in this case they have also to correct the number). The docking analysis would be useful if the selected peptides had been purified and tested for their biological activities.

The conclusion that the selected peptides are effective CE inhibitors is not supported by experimental evidences.

Supplementary figure S4 and S5 are mass spectrum fragmentation, they can be omitted, usually these two figures are the normal output of software for MS/MS experiments.

Minor points

2.4 paragraph contains an error from cut and paste

Line 416 : some words are missing

Reviewer 3 Report

The present study was aimed to assess quinoa as the novel plant-protein derived bioactive peptides with potential anti-hypercholesterolemic activities. The manuscript was well written except some sections (especially discussion and conclusion). The authors should discuss their results with more up-to-date sources and in a more critical and rigorous manner. In the conclusion part of the study, how the findings obtained from this study will be used in the clinical practice, and what are the strengths and limitations of this study should be mentioned. Some places in the tables are shown in yellow and green. This is not very suitable for academic writing. Reference citations and abbrevation usages should be checked.

Round 2

Reviewer 2 Report

The authors did not answer to the following point, or better they wrote that they agree, but no SDS PAGE has been included in the revised version, why? 

Results: the authors should include a SDSPAGE pattern of the quinoa hydrolysates

Moreover the authors wrote that they have removed from the supplementary materials figures S4 and S5, but  the last paragraph in the main text  (line 677-685) has not been changed
